# Adverse events of androgen receptor pathway inhibitors in prostate cancer from real world data

Jongsoo Lee[1], Miho Song[2], Subeen Leem[3], Jong-Yeup Kim[3,4,5], Benjamin I. Chung[6], Sung Ryul Shim[3,4]*, Jae Heon Kim[2]*

1 Department of Urology, Severance Hospital, Yonsei University College of Medicine, Seoul, Republic of Korea, 2 Department of Urology, Soonchunhyang University Seoul Hospital, Soonchunhyang University College of Medicine, Seoul, Republic of Korea, 3 Evidence Based Research Center, Konyang University Hospital, Daejeon, Republic of Korea, 4 Department of Biomedical Informatics, College of Medicine, Konyang University, Daejeon, Republic of Korea, 5 Department of Otorhinolaryngology-Head and Neck Surgery, College of Medicine, Konyang University, Daejeon, Republic of Korea, 6 Department of Urology, Stanford University Medical Center, Stanford, California, United States of America

☯ These authors contributed equally to this work.
* sungryul.shim@gmail.com (SRS), piacekjh@hanmail.net (JHK)

## Abstract

### Purpose

While clinical RCTs have clearly evidenced that the use of androgen receptor pathway inhibitor (ARPI)s for patients with advanced prostate cancer, can significantly delay disease progression, there is insufficient evidence on their safety profiles to warrant their unqualified implementation as the treatment of choice or with which to choose between them. We aim to provide more substantial evidence for adverse event (AE)s of ARPI by analyzing real-world data (RWD) to select optimal ARPI for individual treatment.

### Materials and Methods

We used data from the Food and Drug Administration Adverse Event Reporting System (FAERS) in the US, between April 30, 2014 and April 30, 2024.

### Results

We estimated proportional risk ratio (PRR)s of AEs in the US. We also compared the likelihood of AEs by age, reporter type, and ARPI groups: 1) Group 1, Enzalutamide with other medications; 2) Group 2, Apalutamide with other medications; 3) Group 3, Darolutamide with other medications; 4) Group 4, Abiraterone with other medications; 5) Group 5, Abiraterone + Enzalutamide with other medications. We identified 107,582 AEs among 44,856 US residents who were treated with ARPIs for prostate cancer. By ARPI groups, the AE of GI was the highest in Group 1 and Group 3, and the AE of vascular was the highest in Group 4 and Group 5. In particular, Group 2 showed very

**Data availability statement:** All relevant data are within the paper and its Supporting Information files. (https://www.fda.gov/drugs/fdas-adverse-event-reporting-system-faers/fda-adverse-event-reporting-system-faers-public-dashboard).

**Funding:** This work was supported by Soonchunhyang University Research Fund and National Research Foundation of Korea (NRF) grant funded by the Korea government (MSIT) (2022R1A2C3005586). The funders had no role in study design, data collection and analysis, decision to publish, or preparation of the manuscript.

**Competing interests:** The authors have declared that no competing interests exist.

statistically significantly higher levels of PRR 3.558 (95%CI: 3.489–3.627) of skin-related AE compared to other groups.

## Conclusion

Our study provides important insight that we analyzed RWD and evaluated comparative drug safety across all type of prostate cancer. Although we could not make a conclusion whether which is the safest ARPI, we can suggest that each ARPIs have different types of AEs hence we can use this information during choosing ARPIs for prostate cancer patients.

## Introduction

Prostate cancer is a major global health issue, ranked as the second most commonly diagnosed cancer among men [1,2]. At initial diagnosis, most prostate cancers are localized, yet nearly one-third of these patients eventually experience disease recurrence following initial definitive therapies, such as radical prostatectomy or radiation therapy [3–5]. Although androgen-deprivation therapy (ADT) is the primary treatment for recurrent prostate cancer, resistance commonly develops, transitioning the disease from castration-sensitive prostate cancer (CSPC) to castration-resistant prostate cancer (CRPC), posing significant clinical challenges [6].

Advances in treatment have introduced second-generation androgen receptor pathway inhibitors (ARPIs) as therapeutic options for prostate cancer across different clinical stages [7].

Clinical trials such as ARCHES, ENZAMET, and TITAN have provided strong evidence that adding ARPIs to standard ADT significantly improves overall survival (OS), progression-free survival (PFS), prostate-specific antigen progression-free survival (PSA-PFS), and quality of life (QoL) compared to ADT [8–10]. Historically, nmCRPC represented an area lacking effective treatments, without clear guidelines or consensus on standard care, until the recent FDA approvals of three ARPIs—apalutamide, enzalutamide, and darolutamide—in 2018 and 2019 significantly reshaped this treatment landscape [11,12]. Clinical trials such as SPARTAN, PROSPER, and ARAMIS demonstrated notable improvements in metastasis-free survival (MFS) and OS, supporting their adoption into clinical practice guidelines [8,13,14].

With more treatment options, come more, and often more complex, treatment decisions. Any decisions about drug therapies should involve the patient and healthcare provider, and include all aspects of patient quality of life QoL, not clinical efficacy in isolation. Thus, a particular drug's ability to delay metastases should be weighed against the risks of adverse event (AE)s and other safety aspects, that can have significant effects on patient well-being. Indeed, the clinical efficacy of a particular drug is immaterial if a patient experiences AEs serious enough to necessitate its discontinuation [15]. A holistic approach is required, that considers any concomitant therapies, not necessarily pharmacological or even related to the cancer, including financial

aspects, that have impacts on the daily lives of both patient and caregiver [15,16]. In summary, the choice of ARPI should be individualized, aiming to maximize efficacy while minimizing harm.

The selection of an optimal ARPI for advanced prostate cancer is a critical clinical challenge. Although randomized controlled trial (RCT)s have confirmed the efficacy of various ARPIs, there is a significant lack of comparative data on their adverse effect profiles. Therefore, we aim to provide more substantial evidence for AEs of ARPI by analyzing real world data (RWD), which relied on the AE results of previous RCTs, to help to select optimal ARPI for individual treatment.

The efficacy of ARPIs in delaying advanced prostate cancer progression is also well-documented by RCTs; however, a gap in evidence regarding their safety profiles challenges optimal treatment selection. Recent studies have reported that it is possible to predict the progression of prostate cancer in advance using transcriptomic biomarkers, but it is still not easy to identify side effects using transcriptomic biomarkers [17]. In this research, we aim to clarify the drugs' profiles with respect to AEs, by examining RWD on previous RCTs, and so enable health providers to make more informed choices about the particular ARPI (if any) that would offer most benefits to specific individuals.

## Methods

The study procedures were conducted following the Strengthening the Reporting of Observational Studies in Epidemiology (STROBE) guidelines [18]. This study was exempted by the Institutional Review Board at Konyang University and Soonchunhyang University, as we utilized publicly available, anonymized data, satisfying the requirements for exemption as detailed in the Department of Health and Human Services regulations [19].

### Databases

The FAERS (Food and Drug Administration's Adverse Event Reporting System) database was used to evaluate reported AEs for prostate cancer medication in Enzalutamide, Apalutamide, Darolutamide, and Abiraterone in the United States. FAERS is a voluntary and passive reporting system designed for healthcare providers, pharmaceutical firms, and the public. It allows the submission of reports on AEs, medication errors, and issues related to the quality of drugs and biological products [20]. The database includes details on patient characteristics, medical background, current medications, descriptions of AEs and their outcomes, as well as the origin of the reports. All AEs are classified according to the international Medical Dictionary for Regulatory Activities (MedDRA) [21,22]. These coded terms are structured within a hierarchy of five categories, encompassing both broad (system organ class (SOC) and specific categories (e.g., preferred term (PT)). A PT can encompass various elements such as indicators or symptoms, diagnoses, treatment purposes, lab examinations, interventions, and medical, social, or familial histories [21]. Each FAERS report is associated with one or more PTs, along with an SOC that corresponds to each PT. Each PT is evaluated independently of others.

We collected FAERS data from April 2014 to April 2024 to analyze prostate cancer medication AEs associated with Enzalutamide, Apalutamide, Darolutamide, and Abiraterone authorized in the United States. Information about the survey, including the questionnaires, methodology, and dataset description, can be found on the previously mentioned websites (https://www.fda.gov/drugs/surveillance/fda-adverse-event-reporting-system-faers).

### Adverse events

Since the overall count of AE names were 4,440, which was too many, data preprocessing of all AEs into similar terms and related disease groups was necessary. Of all AE names, more than 100 reported ones were divided into 12 disease groups. According to data preprocessing, a total of 12 disease groups were based on MedDRA concepts at the PT level (Supplemental S1 Table). Two researchers (SRS and JHK) independently reviewed the descriptions in the database to verify the accuracy of the adverse events related to prostate cancer medications AEs. One author (JHK), a specialist in Urology and prostate cancer, validated the identified terms and their groupings. The authors also analyzed all narrative

descriptions of concurrent illnesses and comorbidities in FAERS. In cases of disagreement regarding the descriptions, the final PTs were established through agreement among all investigators.

## Setting and study population

The following five treatment groups were classified according to prostate cancer medication agents (Enzalutamide, Apalutamide, Darolutamide, and Abiraterone) used for prostate cancer; Group 1, Enzalutamide with other medications (excluding other ARPIs); Group 2, Apalutamide with other medications (excluding other ARPIs); Group 3, Darolutamide with other medications (excluding other ARPIs); Group 4, Abiraterone with other medications (excluding other ARPIs); Group 5, Abiraterone + Enzalutamide with other medications (excluding Apalutamide or Darolutamide). Each group by drug was doublet or triplet treatment based on ADT. The age range was categorized into five levels as follows: < 50, 50–59, 60–69, 70–79, and ≥80 years using FAERS.

## Statistical analysis

The percentages of AEs by age and treatments groups were documented. Pearson's chi-squared tests or Fisher's exact tests were conducted to identify statistically significant differences among the categories. The proportional reporting ratio (PRR) is a widely employed approach for evaluating the significance of AEs. It serves as a key metric of disproportionality utilized by the FDA for data analysis within the FAERS database [23,24]. To compute the PRR, the number of the total cases for a specific AE a linked to the prostate cancer treatment group is divided by the number of the same AE for all other treatments in the FAERS database. This calculation is similar to assessing the relative risk of a medication. The formula for PRR is as follows:

$$PRR = \frac{m}{n} / \left[ \frac{M-m}{N-n} \right]$$

(1)

m, indicates the number of cases for the specific AE of the Group 1–5 only.

n, indicates the total number of AEs of the Group 1–5 exclusively.

M, indicates the overall count of cases for the specific AE in the FAERS database.

N, indicates the overall count of all AEs present in the FAERS database.

The PRR is an important metric for assessing the potential relevance of AEs linked to prostate cancer treatments and other drugs. A value of ≥2 suggests a signal that is exceeds than background noise [25–28]. All statistics were conducted as two-tailed, with *p*-values less than 0.05 deemed statistically significant. R version 4.3.1 was utilized for all statistical analyses (R Foundation for Statistical Computing, Vienna, Austria).

## Results

### Characteristics of the study sample

Over the past decade, there were 83,633 patients and 220,064 cases of AEs in the United States that reported AEs associated with prostate cancer medication from April 30, 2014 to April 30, 2024 were collected from the US FAERS database. Since then, missing values of unknown age have been excluded, and the total number of AEs and 44,856 patients in the treatment group 1 to group 5 in this study were 107,582 (Fig 1 and Table 1).

The individual covariates between the treatment groups all showed statistically significant differences. In particular, in the case of Group 3 (79.0%) and Group 5 (90.6%), the rate of reporting serious AEs was relatively higher than that of other groups. In terms of reporting type, Group1 reported relatively higher consumers than other groups (Table 1).

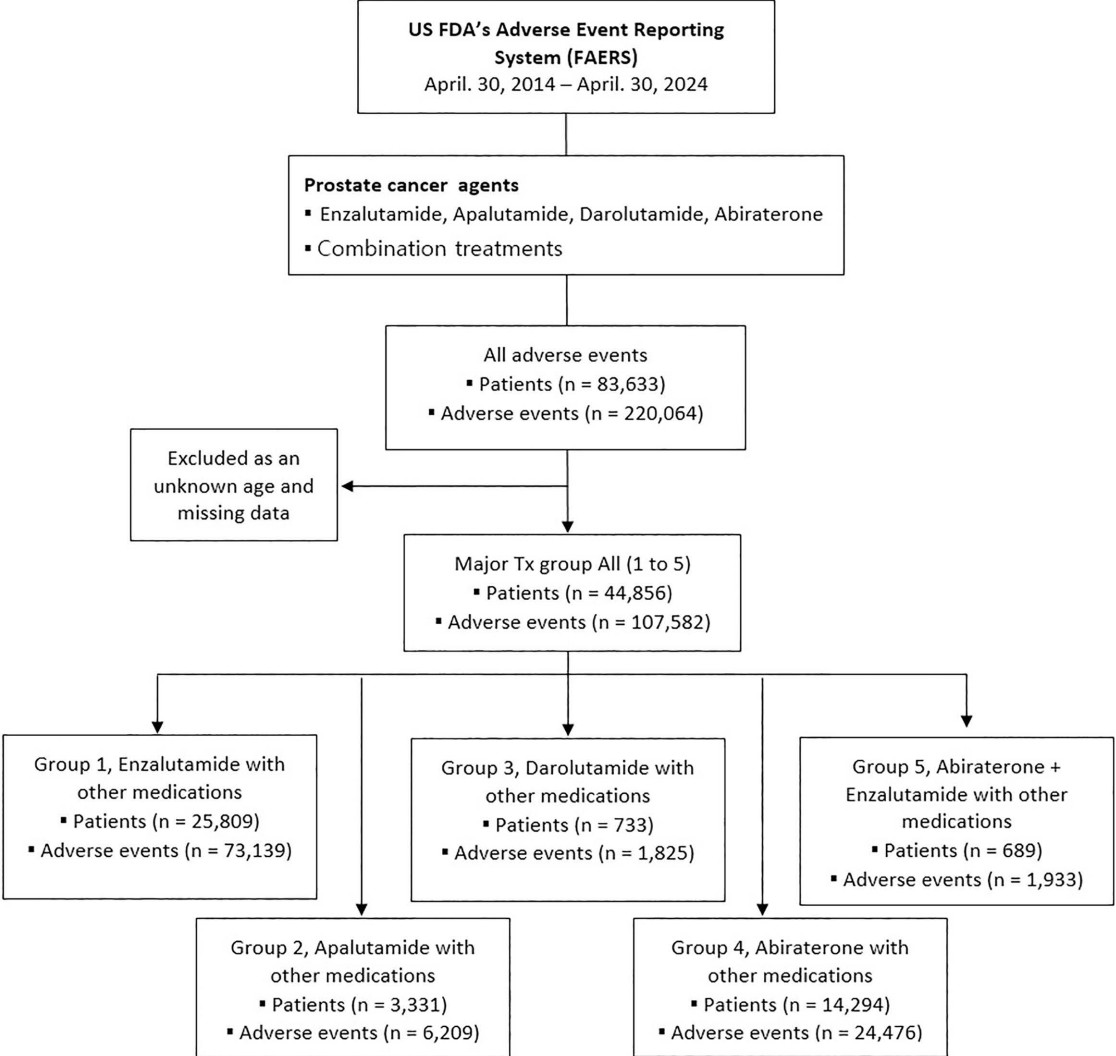

**Fig 1. Flow diagram.** Data are from US FDA's Adverse Event Reporting System (FAERS) through to April 30, 2024. Group 1, Enzalutamide with other medications (excluding other ARPIs); Group 2, Apalutamide with other medications (excluding other ARPIs); Group 3, Darolutamide with other medications (excluding other ARPIs); Group 4, Abiraterone with other medications (excluding other ARPIs); Group 5, Abiraterone + Enzalutamide with other medications (excluding Apalutamide or Darolutamide). P-value, Chi-square test or Fisher's exact test. Missing values removed.

## Comparison of AEs by age group

Looking at the frequency of detailed 12 disease groups by age group, the highest AEs in all age groups were found to lack of efficacy. When looking at the detailed diseases excluding lack of efficacy and general complications, those under the age of 50, 50–59, and 60s complained of the AEs of musculoskeletal and gastrointestinal diseases the most. Those in their 70s and over 80 years of age complained of AEs of vascular disease and gastrointestinal disease the most (Table 2).

## Comparison of AEs by treatment group

Looking at the frequency of each treatment group of 12 disease groups, the highest AE in all treatment groups was also found to lack of efficacy. Looking at the detailed diseases excluding lack of efficacy and general complications, Group 1

**Table 1. Characteristics of adverse events of prostate cancer treatments in the United States.**

| Characteristics | | Treatment groups | | | | | |
|---|---|---|---|---|---|---|---|
| | | Group 1 (N = 25,809) | Group 2 (N = 3,331) | Group 3 (N = 733) | Group 4 (N = 14,294) | Group 5 (N = 689) | *P*-value |
| Serious | | | | | | | <0.001 |
| | Serious | 17355 (67.2%) | 2221 (66.7%) | 579 (79.0%) | 9523 (66.6%) | 624 (90.6%) | |
| | Non-serious | 8454 (32.8%) | 1110 (33.3%) | 154 (21.0%) | 4771 (33.4%) | 65 (9.4%) | |
| Case.Priority | | | | | | | <0.001 |
| | Direct | 845 (3.3%) | 955 (28.7%) | 20 (2.7%) | 5654 (39.6%) | 5 (0.7%) | |
| | Expedited | 14877 (57.6%) | 1722 (51.7%) | 559 (76.3%) | 6236 (43.6%) | 529 (76.8%) | |
| | Non-Expedited | 10087 (39.1%) | 654 (19.6%) | 154 (21.0%) | 2404 (16.8%) | 155 (22.5%) | |
| Reporter.Type | | | | | | | <0.001 |
| | Consumer | 18089 (70.1%) | 1032 (31.0%) | 368 (50.2%) | 3895 (27.2%) | 155 (22.5%) | |
| | Healthcare Professional | 7586 (29.4%) | 2019 (60.6%) | 365 (49.8%) | 9591 (67.1%) | 530 (76.9%) | |
| | Not Specified | 134 (0.5%) | 280 (8.4%) | 0 (0.0%) | 808 (5.7%) | 4 (0.6%) | |
| Age | | | | | | | <0.001 |
| | <50 | 127 (0.5%) | 9 (0.3%) | 8 (1.1%) | 90 (0.6%) | 7 (1.0%) | |
| | 50-59 | 1248 (4.8%) | 107 (3.2%) | 46 (6.3%) | 765 (5.4%) | 37 (5.4%) | |
| | 60-69 | 5140 (19.9%) | 640 (19.2%) | 156 (21.3%) | 3267 (22.9%) | 165 (23.9%) | |
| | 70-79 | 9525 (36.9%) | 1311 (39.4%) | 252 (34.4%) | 5280 (36.9%) | 283 (41.1%) | |
| | ≥80 | 9769 (37.9%) | 1264 (37.9%) | 271 (37.0%) | 4892 (34.2%) | 197 (28.6%) | |
| | | | | | | | |

Note: Data are from US FDA's Adverse Event Reporting System (FAERS) through to April 30, 2024. Group 1, Enzalutamide with other medications (excluding other ARPIs); Group 2, Apalutamide with other medications (excluding other ARPIs); Group 3, Darolutamide with other medications (excluding other ARPIs); Group 4, Abiraterone with other medications (excluding other ARPIs); Group 5, Abiraterone + Enzalutamide with other medications (excluding Apalutamide or Darolutamide). P-value, Chi-square test or Fisher's exact test. Missing values removed.

showed the order of gastrointestinal (11.4%), CNS (10.5%), and musculoskeletal (9.7%); Group 2 showed the order of skin (12.5%), vascular (10.9%), and CNS (7.9%); Group 3 showed the order of gastrointestinal (10.6%), vascular (10.4%), and musculoskeletal (10.0%); Group 4 showed the order of vascular (12.0%) and gastrointestinal (10.8%); Group 5 showed the order of vascular (12.8%), gastrointestinal (10.9%), and musculoskeletal (10.4%) (Table 3).

### Proportional reporting ratio by treatment group

As a result of analyzing all samples including Group 1 to Group 5, infection 2.280 (95% confidential interval (CI), 2.215 to 2.345), kidney/urology 2.215 (95% CI, 2.166 to 2.264), and respiratory 2.156 (95% CI, 2.103 to 2.209) showed the top three highest PRRs (Fig 2). In the case of Group 2, PRR was shown in the order of skin 3.558 (95% CI, 3.489 to 3.627), infection 1.937 (95% CI, 1.802 to 2.072), and endocrine 1.704 (95% CI, 1.582 to 1.826). In particular, Group 2 showed significantly higher PRR for skin-related AEs than other treatment and disease groups. Detailed disease-specific PRRs by treatment group can be found in the supplementary materials (Supplemental S2–S7 Tables)

### Discussion

Our study provides important insight that we analyzed real word database and evaluated comparative drug safety across all type of prostate cancer. Although we could not make a conclusion whether which is the safest ARPI for prostate cancer, we can suggest that each ARPIs have different types of AEs hence we can use this information during choosing ARPIs for prostate cancer patients.

**Table 2. Adverse Events of prostate cancer treatments in the United States by age group.**

| Symptoms | Age groups | | | | | P-value |
|---|---|---|---|---|---|---|
| | **<50** | **50-59** | **60-69** | **70-79** | **>80** | |
| | **Frequency (%)** | **Frequency (%)** | **Frequency (%)** | **Frequency (%)** | **Frequency (%)** | |
| Lack of efficacy | 104 (16.99) | 953 (18.77) | 3,761 (17.14) | 6,303 (15.66) | 7,094 (17.87) | <0.001 |
| General complications | 70 (11.44) | 536 (10.56) | 2,508 (11.43) | 5,203 (12.93) | 5,082 (12.8) | <0.001 |
| Infection | 15 (2.45) | 118 (2.32) | 535 (2.44) | 1,064 (2.64) | 1,062 (2.68) | 0.283 |
| CNS | 42 (6.86) | 443 (8.73) | 1,932 (8.8) | 3,680 (9.14) | 3,803 (9.58) | 0.002 |
| OPH/ENT | 11 (1.8) | 99 (1.95) | 556 (2.53) | 1,358 (3.37) | 1,202 (3.03) | <0.001 |
| Respiratory | 18 (2.94) | 168 (3.31) | 673 (3.07) | 1,589 (3.95) | 1,750 (4.41) | <0.001 |
| Musculoskeletal | 66 (10.78) | 537 (10.58) | 2,079 (9.47) | 3,543 (8.8) | 3,485 (8.78) | <0.001 |
| Vascular | 48 (7.84) | 409 (8.06) | 2,045 (9.32) | 4,164 (10.34) | 3,972 (10.01) | <0.001 |
| Endocrine | 21 (3.43) | 211 (4.16) | 921 (4.2) | 1,254 (3.12) | 739 (1.86) | <0.001 |
| Gastro intestinal | 82 (13.4) | 564 (11.11) | 2,570 (11.71) | 4,566 (11.34) | 4,118 (10.37) | <0.001 |
| Kidney/Urology | 16 (2.61) | 224 (4.41) | 1,078 (4.91) | 1,901 (4.72) | 1,808 (4.55) | 0.028 |
| Skin | 47 (7.68) | 291 (5.73) | 1,269 (5.78) | 2,071 (5.15) | 1,558 (3.92) | <0.001 |
| Others | 72 (11.76) | 523 (10.3) | 2,019 (9.2) | 3,556 (8.83) | 4,023 (10.13) | <0.001 |
| Total | 612 | 5,076 | 21,946 | 40,252 | 39,696 | |

Note: Data are from US FDA's Adverse Event Reporting System (FAERS) through to April 30, 2024. Group 1, Enzalutamide with other medications (excluding other ARPIs); Group 2, Apalutamide with other medications (excluding other ARPIs); Group 3, Darolutamide with other medications (excluding other ARPIs); Group 4, Abiraterone with other medications (excluding other ARPIs); Group 5, Abiraterone + Enzalutamide with other medications (excluding Apalutamide or Darolutamide). P-value, Chi-square test or Fisher's exact test. Missing values removed. Allow more than one adverse events calculation per patient.

Through analyses of RWD, we compared the safety of the three second generation ARPIs in the treatment of all types of prostate cancer. The study produced important insights about the types and reporting frequency of AEs associated with each ARPI, although recommending one as the safest is not possible. The measure we adopted in the research was the PRR, which compares the frequencies of reported AEs associated with particular ARPIs to the expected frequencies for those AEs if there were no drug-AE associations. Thus, for any specific AE, the measure compares a single drug to all other drugs. Recent met-analysis study showed that combined treatments result in better survival than does ADT alone [29]. About the side effects of ARPI, one recent meta-analysis showed that darolutamide had a toxicity profile comparable to placebo with the exception of bone fractures [30]. In the absence of head-to-head comparison studies between the different ARPIs, the results of this meta-analysis suggested a preferred use of darolutamide in the approved.

According to the characteristic PRR ranking for each drug, Enzalutamide showed OPH/ENT symptoms, CNS symptoms, and musculo-skeletal symptoms (highest in order). Apalutamide showed skin symptoms, infectious symptoms, and endocrine symptoms (highest in order). Darolutamide showed skin symptoms, nephrologic and urologic symptoms, and infectious symptoms (highest in order). Abiraterone showed nephrologic and urologic symptoms, infectious symptoms, and respiratory symptoms (highest in order). Combination of abiraterone and enzalutamide showed nephrologic and urologic symptoms, and respiratory symptoms (highest in order). ARPI should be selected according to the characteristics of individual patients.

Recent network meta-analysis (NMA) study concluded that abiraterone acetate and apalutamide offered the largest overall survival benefits among mCSPC treatments, with relatively low serious AE risks [28]. In other words, clinical trial data suggest that second-generation ARPIs can substantially improve survival outcomes without a dramatic trade-off in safety, especially when compared to older treatments like chemotherapy. Although indirect comparisons cannot substitute for direct head-to-head evidence, the understanding from the studies collectively reinforces that ARPIs possess distinct safety profiles with specific advantages and limitations.

**Table 3. Adverse Events of prostate cancer treatments in the United States by treatment group.**

| Symptoms | Treatment groups | | | | | P-value |
|---|---|---|---|---|---|---|
| | Group 1 | Group 2 | Group 3 | Group 4 | Group 5 | |
| | Frequency (%) | Frequency (%) | Frequency (%) | Frequency (%) | Frequency (%) | |
| Lack of efficacy | 12,251 (16.75) | 783 (12.61) | 138 (7.56) | 4,720 (19.28) | 323 (16.71) | <0.001 |
| General complications | 10,329 (14.12) | 678 (10.92) | 225 (12.33) | 1,977 (8.08) | 190 (9.83) | <0.001 |
| Infection | 1,518 (2.08) | 217 (3.49) | 57 (3.12) | 929 (3.8) | 73 (3.78) | <0.001 |
| CNS | 7,677 (10.5) | 496 (7.99) | 162 (8.88) | 1,403 (5.73) | 162 (8.38) | <0.001 |
| OPH/ENT | 2,574 (3.52) | 171 (2.75) | 39 (2.14) | 412 (1.68) | 30 (1.55) | <0.001 |
| Respiratory | 2,485 (3.4) | 239 (3.85) | 82 (4.49) | 1,274 (5.21) | 118 (6.1) | <0.001 |
| Musculoskeletal | 7,103 (9.71) | 450 (7.25) | 183 (10.03) | 1,774 (7.25) | 200 (10.35) | <0.001 |
| Vascular | 6,573 (8.99) | 682 (10.98) | 189 (10.36) | 2,946 (12.04) | 248 (12.83) | <0.001 |
| Endocrine | 2,039 (2.79) | 258 (4.16) | 51 (2.79) | 754 (3.08) | 44 (2.28) | <0.001 |
| Gastro intestinal | 8,339 (11.4) | 523 (8.42) | 193 (10.58) | 2,633 (10.76) | 212 (10.97) | <0.001 |
| Kidney/Urology | 2,736 (3.74) | 270 (4.35) | 111 (6.08) | 1,765 (7.21) | 145 (7.5) | <0.001 |
| Skin | 3,487 (4.77) | 776 (12.5) | 139 (7.62) | 763 (3.12) | 71 (3.67) | <0.001 |
| Others | 6,028 (8.24) | 666 (10.73) | 256 (14.03) | 3,126 (12.77) | 117 (6.05) | <0.001 |
| Total | 73,139 | 6,209 | 1,825 | 24,476 | 1,933 | |

Note: Data are from US FDA's Adverse Event Reporting System (FAERS) through to April 30, 2024. Group 1, Enzalutamide with other medications (excluding other ARPIs); Group 2, Apalutamide with other medications (excluding other ARPIs); Group 3, Darolutamide with other medications (excluding other ARPIs); Group 4, Abiraterone with other medications (excluding other ARPIs); Group 5, Abiraterone + Enzalutamide with other medications (excluding Apalutamide or Darolutamide). P-value, Chi-square test or Fisher's exact test. Missing values removed. Allow more than one adverse events calculation per patient.

A systematic review by Wang et al [28], found considerably greater serious AEs (odds ratio, 23.72; 95% CI, 13.37–45.15) with docetaxel, moderately increased serious AEs (odds ratio, 1.42; 95% CI, 1.10–1.83) with abiraterone acetate, while none of the other drugs showed any increased risks. When compared to enzalutamide, apalutamide was found to have lower risks for all SAEs and the non-SAEs fatigue, hypertension, falls and seizures by Santoni et al. [31] although they observed less risk of developing anemia with enzalutamide. Indirect comparisons have shown darolutimade to have significantly better safety profiles than apalutamide and enzalutamide, [32] with both of the latter drugs linked to higher risks of AEs involving the CNS, than darolutimade.

Mechanistic differences between the drugs likely contribute to their divergent safety profiles. All three second-generation ARPIs share a similar mechanism of action, but they differ in molecular structure and pharmacokinetics. Notably, darolutamide has a relatively large molecular size and polar structure that limits its penetration across the blood–brain barrier. This property is thought to underlie the lower frequency of CNS-related adverse events seen with darolutamide, in contrast to apalutamide and enzalutamide which more readily enter the central nervous system [33,34]. Halabi et al, for instance, found darolutamide conferred significantly lower absolute risks of falls and fractures than either apalutamide or enzalutamide, as well as a lower risk of cognitive impairment than enzalutamide [32]. In addition, due to the short observation period, darolutamide must be careful in interpreting the side effects involved, and more accumulated data must be completed and then reanalyzed.

Similarly, an indirect risk-difference analysis by Di Nunno et al. observed a higher fracture risk with apalutamide but no increased fracture risk with darolutamide (versus placebo) [35]. Consistent with these findings, the incidence of falls and fractures with apalutamide and enzalutamide has been notably high, to the point that prescribing information for both agents recommends evaluating patients for fall and fracture risk prior to initiating therapy [36,37]. While these indirect comparisons cannot replace direct head-to-head evidence, they collectively reinforce the notion that each ARPI has

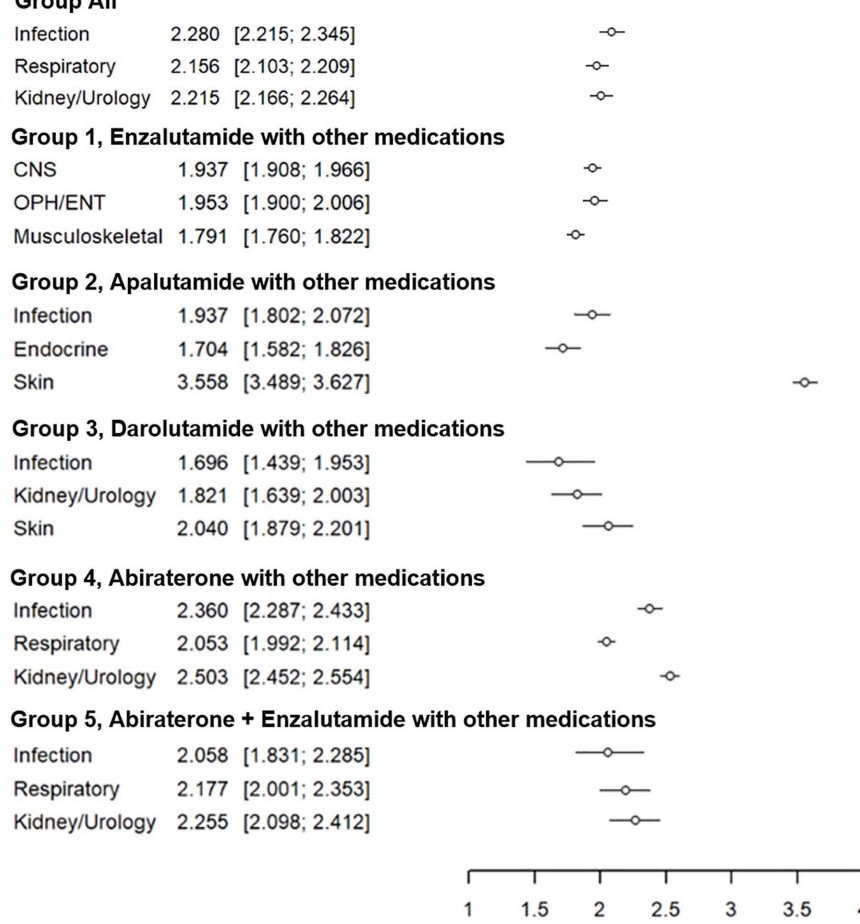

**Fig 2. Proportional reporting ratios and 95% confidential intervals in Group 1 to Group 5.** Data are from US FDA's Adverse Event Reporting System (FAERS) through to April 30, 2024. Group 1, Enzalutamide with other medications (excluding other ARPIs); Group 2, Apalutamide with other medications (excluding other ARPIs); Group 3, Darolutamide with other medications (excluding other ARPIs); Group 4, Abiraterone with other medications (excluding other ARPIs); Group 5, Abiraterone + Enzalutamide with other medications (excluding Apalutamide or Darolutamide). PRR, proportional reporting ratio. Top 3 statistically high risk adverse events by group.

distinct side-effect liabilities – with darolutamide often emerging as the most benign in terms of CNS and musculoskeletal side effects, and enzalutamide and apalutamide sharing certain toxicities [38].

A recent NMA and systematic review of thirteen, randomized, double-blinded trials which studied abiraterone, apalutamide, darolutamide or enzalutamide on all prostate cancer types, carried out by Cao et al [39], found only two significant differences in their side-effects – abiraterone's lower risk of musculoskeletal AEs and enzalutamide's higher risk for cardiovascular AEs. With regard to any AEs (not just SAEs), the RRs to placebo were lowest for darolutamide 1.17 (95% CI, 0.83–1.65), followed by abiraterone 1.19 (95% CI, 0.98–1.44), apalutamide 1.20 (95% CI, 0.95–1.53), and highest for enzalutamide 1.30 (95% CI, 1.13–1.50). In the three primary measures, PFS, PSA-PFS, and MFS, darolutamide was less efficacious than the other ARSIs and no differences were found between apalutamide and enzalutamide. Other similar meta-analyses have suggested darolutamide likely to be better tolerated than other ARSIs as evidenced by fewer serious AEs (grades 3+) and less toxicity [40,41], and relative equivalence was observed in enzalutamide and apalutamide [42].

Current knowledge of ARPI side effects relies on RCTs and related meta-analyses. However, significant heterogeneity among these studies leads to inconsistent conclusions about adverse events. Large-scale observational studies using

RWD are needed to overcome this limitation, but such research has been absent. Our study is the first to address this gap by analyzing ARPI side effects using a large RWD cohort.

Our study has several limitations, first, it is a cross-sectional study, so we cannot obtain time dependent information, and we cannot obtain information on the stage and clinical situation of the individual. In addition, reports of AEs may include ADT in combination with or in combination with other chemotherapy regimen, making it difficult to distinguish whether those reported AEs are solely due to ARPIs. Second, even in this FAERS data, there is a significant number of participants who did not report their age and gender, which appears to be a significant bias. However, FAERS is a large big data set and we believe that the PRR for each of the individual groups of ARPI (Enzalutamide, Apalutamide, Darolutamide, Abiraterone, and Abiraterone with Enzalutamide), which is our primary interest, is likely to remain constant with a constant effect size because of the large number of missing participants. In addition, we can assume that the missingness also occurred in all 12 disease groups in this study, so it is judged to be a non-differential misclassification error, and the direction is not always towards the null value. Therefore, we can maintain confidence in this study. Third, the inability to control for potential confounders limits its applicability to real-world practice. Unfortunately, FAERS does not provide detailed exact covariates (comorbidities, concurrent medications, disease stage, etc.) for individual participants, so we tried to control for these as much as possible through stratification by age, gender, and treatment group. The findings should therefore be interpreted as hypothesis-generating and descriptive of reporting trends, rather than definitive comparative risk quantifications as it is primarily stemming from the FAERS data.

Future research might also explore pharmacogenomic or molecular markers that predispose patients to certain adverse events, enabling even more tailored ARPI therapy. By deepening our understanding of both the benefits and risks of each ARPI in routine practice, we can optimize prostate cancer treatment strategies to improve survival rates and QoL while minimizing harm.

## Supporting information

**S1 Table. The adverse events of prostate cancer medication were based on Medical Dictionary for Regulatory Activities (MedDRA) concepts at the preferred term (PT) level.**
(PDF)

**S2 Table. Proportional reporting ratios in Group 1 to Group 5.**
(PDF)

**S3 Table. Proportional reporting ratios in Group 1 (Enzalutamide with other medications (excluding other ARPIs).**
(PDF)

**S4 Table. Proportional reporting ratios in Group 2 (Apalutamide with other medications (excluding other ARPIs).**
(PDF)

**S5 Table. Proportional reporting ratios in Group 3 (Darolutamide with other medications (excluding other ARPIs).**
(PDF)

**S6 Table. Proportional reporting ratios in Group 4 (Abiraterone with other medications (excluding other ARPIs).**
(PDF)

**S7 Table. Proportional reporting ratios in Group 5 (Abiraterone+Enzalutamide with other medications (excluding Apalutamide or Darolutamide).**
(PDF)

**S8 Table. STROBE Statement—checklist of items that should be included in reports of observational studies.**

(PDF)

## Author contributions

**Conceptualization:** Sung Ryul Shim, Jae Heon Kim.

**Data curation:** Jongsoo Lee, Miho Song, Subeen Leem, Sung Ryul Shim.

**Formal analysis:** Jongsoo Lee, Miho Song, Subeen Leem, Jong-Yeup Kim, Benjamin I. Chung, Sung Ryul Shim, Jae Heon Kim.

**Funding acquisition:** Jae Heon Kim.

**Investigation:** Jongsoo Lee, Miho Song, Jong-Yeup Kim, Benjamin I. Chung, Sung Ryul Shim, Jae Heon Kim.

**Methodology:** Subeen Leem, Sung Ryul Shim.

**Supervision:** Sung Ryul Shim, Jae Heon Kim.

**Validation:** Jong-Yeup Kim, Sung Ryul Shim.

**Visualization:** Sung Ryul Shim.

**Writing – original draft:** Benjamin I. Chung, Sung Ryul Shim, Jae Heon Kim.

**Writing – review & editing:** Jongsoo Lee, Miho Song, Subeen Leem, Jong-Yeup Kim, Benjamin I. Chung, Sung Ryul Shim, Jae Heon Kim.

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
