## [Decision Letter · Decision Letter 0]

5 May 2025

Dear Dr. SHIM,

Thank you for submitting your manuscript to PLOS ONE. After careful consideration, we feel that it has merit but does not fully meet PLOS ONE’s publication criteria as it currently stands. Therefore, we invite you to submit a revised version of the manuscript that addresses the points raised during the review process.

We look forward to receiving your revised manuscript.

Kind regards,

Xing-Xiong An, M.D.

Academic Editor

PLOS ONE

Journal Requirements:

2. Thank you for stating the following financial disclosure: [This work was supported by Soonchunhyang University Research Fund. National Research Foundation of Korea (NRF) grant funded by the Korea government (MSIT) (2022R1A2C3005586)].  

3. Thank you for uploading your study's underlying data set. Unfortunately, the repository you have noted in your Data Availability statement does not qualify as an acceptable data repository according to PLOS's standards.

Reviewers' comments:

Reviewer's Responses to Questions

**Comments to the Author**

1. Is the manuscript technically sound, and do the data support the conclusions?

Reviewer #1: Yes

Reviewer #2: Yes

Reviewer #3: Yes

Reviewer #4: Yes

2. Has the statistical analysis been performed appropriately and rigorously?

Reviewer #1: No

Reviewer #2: I Don't Know

Reviewer #3: Yes

Reviewer #4: Yes

3. Have the authors made all data underlying the findings in their manuscript fully available?

Reviewer #1: Yes

Reviewer #2: Yes

Reviewer #3: Yes

Reviewer #4: Yes

4. Is the manuscript presented in an intelligible fashion and written in standard English?

Reviewer #1: Yes

Reviewer #2: No

Reviewer #3: Yes

Reviewer #4: Yes

Reviewer #1: Interesting topic.

By addressing these points, the study will be clearer, more rigorous, and clinically relevant.

1. Context on ARPI Side Effects

You would better cite recent clinical meta-analyses (e.g., PMIDs: 39223303, 38097723) to address the evidence gap on ARPI side effects.

Highlight the need for real-world data and consider discussing why different ARPIs have varied side effect profiles.

2. Treatment Group Rationale

Please clarify the logic behind grouping ARPIs for a clearer understanding, especially for non-specialist readers.

Expand the Methods section to better explain groupings and how they reflect real-world prescribing practices.

3. Attribution of Side Effects to Specific ARPIs

You have mentioned that combination therapies make it hard to attribute side effects to a single drug.

Suggest further analysis to separate combination treatments and emphasize that PRRs can't establish causality.

4. Generalizability and Regional Limitations

Acknowledge that while flutamide is FDA-approved, it was not included in the study. This exclusion may limit the generalizability and the findings may not apply globally, particularly for regions still using non-FDA-approve or other older therapies.

5. Confounder Adjustment

Adjust for potential confounders (e.g., age, comorbidities) to ensure accurate results.

6. Reporting Biases in FAERS

Address biases in FAERS data (underreporting, overreporting) and how they impact the results.

7. Temporal Bias

Consider the shorter reporting window for newer drugs like darolutamide, which could skew AE comparisons.

8. Clinical Actionability

Suggest how clinicians might use the findings in practice, such as avoiding apalutamide for patients with skin issues.

9. Sample Size Concerns

Note that smaller groups (e.g., Group 3, n=733) may result in less stable PRR estimates. Consider confidence interval overlap in comparisons.

10. Language and Clarity

Recommend improving clarity and correcting grammar (e.g., "instalement" to "implementation").

Reviewer #2: This study provides a valuable contribution to the field of prostate cancer treatment by analyzing real-world data (RWD) from the FAERS database to evaluate the adverse events (AEs) associated with androgen receptor pathway inhibitors (ARPIs). The large sample size and comprehensive analysis of AEs across different ARPI groups offer important insights into their safety profiles. The use of proportional reporting ratios (PRRs) to quantify AE risks is methodologically sound, and the findings highlight the need for individualized treatment decisions based on AE profiles. The study addresses a critical gap in the literature by leveraging RWD to complement clinical trial evidence, making it highly relevant for clinicians and researchers. The manuscript is promising but requires revisions to address these issues before acceptance.

Comments for Revision:

1.

The abstract mentions data collection from May 30, 2014, to May 30, 2024, but the results section states April 30, 2014, to April 30, 2024. This discrepancy needs correction for consistency.

2.

The term "ANY" in the treatment groups (e.g., Enzalutamide+ANY) is ambiguous. Specify whether it refers to concomitant medications, other ARPIs, or additional therapies to avoid confusion.

3.

The rationale for excluding cases with unknown age or missing data is not provided. Explain how this exclusion might impact the representativeness of the results.

4.

The study does not account for potential confounders such as comorbidities, concurrent medications, or disease stage. Discuss how these factors might influence AE reporting and interpretations.

5.

Provide more details on the statistical methods used, such as how PRRs were calculated and adjusted for multiple comparisons. Clarify whether any corrections (e.g., Bonferroni) were applied to mitigate false positives.

6.

The passive and voluntary nature of FAERS reporting can lead to underreporting or bias. Elaborate on how these limitations might affect the study's conclusions and suggest ways to mitigate them.

7.

While PRRs highlight statistical associations, their clinical significance is unclear. Discuss whether the observed PRRs translate to meaningful differences in patient outcomes or treatment decisions.

8.

The criteria for grouping AEs into 12 categories (e.g., "vascular," "CNS") are not detailed. Provide a clear methodology or reference for how these groupings were determined.

9.

The study compares ARPIs indirectly but lacks direct head-to-head comparisons. Acknowledge this limitation and discuss how it affects the interpretation of relative safety.

10.

Although IRB approval was waived, briefly discuss ethical implications of using publicly available data without patient consent, especially regarding privacy and data anonymization.

Reviewer #3: -In this study, the authors analyzed the adverse effects of prostate cancer medication agents (Enzalutamide, Apalutamide, Darolutamide, and Abiraterone) in prostate cancer patients from real-world data.

-ARPIs are known to be effective in advanced prostate cancer treatments. It is important to evaluate the adverse events of treatments together with the patient and decide on the treatment option. Therefore, I think this type of research is valuable.

- The adverse effects of treatments applied to patients can directly affect the quality of life. My suggestion to the authors; Literature data on the quality of life of patients receiving ARPI treatment can be briefly commented on in the discussion section.

Reviewer #4: This manuscript is a retrospective observational study analyzing real-world adverse events (AEs) associated with androgen receptor pathway inhibitors (ARPIs) in prostate cancer patients using the FDA Adverse Event Reporting System (FAERS) database. The study provides valuable insights into the comparative safety profiles of different ARPIs, which can aid clinicians in personalized treatment decisions.

-Is there anyway can the authors handle the missing data instead of exclufing them? please justify why did you remove them instead of missing value imputation technique.

-Analyze AEs by cancer stage or prior treatment lines. Highlight in the introduction stage prostate cnacer analysis prevouis work such as PMID: 30890858 or similar work.

-Expand on how findings should guide ARPI selection (e.g., avoiding apalutamide in patients with skin conditions).

-please highlight the limitation of this work.

**Do you want your identity to be public for this peer review?** For information about this choice, including consent withdrawal, please see our Privacy Policy

Reviewer #1: **Yes: ** Mohammad Reza Fattahi

Reviewer #2: **Yes: ** Mohammad Ebrahimnezhad

Reviewer #3: No

Reviewer #4: **Yes: ** Abedalrhman Alkhateeb

---

## [Author Response · Author response to Decision Letter 1]

4 Jul 2025

Please check the attached file "Response to Reviewers.docx"

---

## [Decision Letter · Decision Letter 1]

11 Aug 2025

Dear Dr. SHIM,

Thank you for submitting your manuscript to PLOS ONE. After careful consideration, we feel that it has merit but does not fully meet PLOS ONE’s publication criteria as it currently stands. Therefore, we invite you to submit a revised version of the manuscript that addresses the points raised during the review process.

We look forward to receiving your revised manuscript.

Kind regards,

Xing-Xiong An, M.D.

Academic Editor

PLOS ONE

Journal Requirements:

Reviewers' comments:

Reviewer's Responses to Questions

**Comments to the Author**

Reviewer #1: (No Response)

Reviewer #3: All comments have been addressed

Reviewer #4: All comments have been addressed

2. Is the manuscript technically sound, and do the data support the conclusions?

Reviewer #1: Yes

Reviewer #3: Yes

Reviewer #4: Yes

3. Has the statistical analysis been performed appropriately and rigorously?

Reviewer #1: Yes

Reviewer #3: Yes

Reviewer #4: Yes

4. Have the authors made all data underlying the findings in their manuscript fully available?

Reviewer #1: Yes

Reviewer #3: Yes

Reviewer #4: Yes

5. Is the manuscript presented in an intelligible fashion and written in standard English?

Reviewer #1: Yes

Reviewer #3: Yes

Reviewer #4: Yes

Reviewer #1: Thank you for the thoughtful revisions and the improvements made to the manuscript. You've addressed several important points, but there are still a few areas that could use a bit more clarification or refinement:

There are a few instances of repetitive content between nearby sentences in both the Introduction and Discussion (e.g., lines 102–105 & 108–111; 309–312 & 316–319). Consider tightening these sections to improve clarity and flow.

In line 167, "three levels" may be a typo based on the context of the results — consider double-checking this.

I’m a bit unclear about the "All treatments" group in Table S2 — does it refer to all AE reports related to ARPIs (including excluded ones), or to all prostate cancer treatments in general? Based on Figure 1, it seems to be the former. If that's the case, the consistently higher PRRs in the included cases compared to all cases (included + excluded) suggest that patients with a higher burden of AEs were retained. This could challenge the authors’ claim that excluded cases had minimal impact on overall results. Even though PRRs aren’t directly comparable across AE categories, it's notable that groups like “Infection” (PRR: 2.28) appear overrepresented compared to categories like “Endocrine” (PRR: 1.41). Some clarification would be helpful here.

A brief explanation of how the confidence intervals were calculated (e.g., method or formula) in the Methods section would be helpful for clarity.

It would be helpful to explain the rationale behind the AE classification—beyond the process outlined in Table S1. Was an established or previously used method followed? For instance, categorizing "pneumonia" under Infection rather than Respiratory could be debated, so some clarification on the reasoning would strengthen the approach.

I noticed that although the use of ARPIs hasn’t declined in recent years, the number of reported AEs has gone down, according to FAERS data. I understand this could be due to many factors, but could changes in treatment strategies—like using lower doses of abiraterone or prescribing ARPIs earlier in the disease—be playing a role? If so, splitting the study period into earlier vs. later years might help clarify whether such shifts are contributing to the trend.

In line 239, the term "prevalence" may not be the most accurate choice—perhaps a more precise term like “reporting frequency” or “proportion of reports” would better reflect the data.

Given the nature of the data, it’s important to note that a higher PRR for a specific AE—like skin complications—doesn’t necessarily mean the drug causes more of those events compared to others. I believe this limitation should be acknowledged more clearly in the manuscript.

Since the combination of abiraterone and enzalutamide is not a routine practice with established benefits, it would be helpful to clarify why this specific pairing was chosen over other possible combinations. Also, was this use concurrent or sequential? Some explanation would strengthen the rationale.

I recommend adjusting for other medications used in combination therapies, such as ADT or docetaxel. Have the authors considered stratifying groups based on whether these agents were co-administered? I understand sample size may be a limitation, but exploring this could add valuable context.

The discussion would benefit from adding more clinically relevant implications and clearer take-home messages for physicians—while still emphasizing appropriate caution given the limitations of the data source.

The discussion would be stronger with a clearer acknowledgment of the potential impact of underreporting and overreporting inherent to the FAERS dataset, as this is a key limitation in interpreting the findings.

Reviewer #3: I thank the authors for their revisions to the article. I recommend that all previously suggested revisions be addressed: ‘‘The adverse effects of treatments applied to patients can directly affect the quality of life. My suggestion to the authors; Literature data on the quality of life of patients receiving ARPI treatment can be briefly commented on in the discussion section.’’

Reviewer #4: The authors addressed the reviewers' comments. The paper is in a very good shape. I endorse it for publication.

**Do you want your identity to be public for this peer review?** For information about this choice, including consent withdrawal, please see our Privacy Policy

Reviewer #1: **Yes: ** Mohammad Reza Fattahi

Reviewer #3: No

Reviewer #4: **Yes: ** Abedalrhman Alkhateeb

---

## [Author Response · Author response to Decision Letter 2]

21 Sep 2025

We sincerely thank the editor and reviewers for their thoughtful and valuable comments.

In attached file of "Response to Reviewers 250922.docx", the authors have answered the reviewers' questions to the best of our ability and hope that this study will be a step forward in the scientific understanding of the adverse effects of ARPIs in prostate cancer with RWD.

---

## [Decision Letter · Decision Letter 2]

13 Oct 2025

Adverse events of androgen receptor pathway inhibitors in prostate cancer from real world data

PONE-D-25-09399R2

Dear Dr. SHIM,

We’re pleased to inform you that your manuscript has been judged scientifically suitable for publication and will be formally accepted for publication once it meets all outstanding technical requirements.

Kind regards,

Xing-Xiong An, M.D.

Academic Editor

PLOS ONE

Additional Editor Comments (optional):

Reviewers' comments:

Reviewer's Responses to Questions

**Comments to the Author**

Reviewer #3: All comments have been addressed

2. Is the manuscript technically sound, and do the data support the conclusions?

Reviewer #3: Yes

3. Has the statistical analysis been performed appropriately and rigorously?

Reviewer #3: Yes

4. Have the authors made all data underlying the findings in their manuscript fully available?

Reviewer #3: Yes

5. Is the manuscript presented in an intelligible fashion and written in standard English?

Reviewer #3: Yes

Reviewer #3: I want to thank the authors for their thorough consideration and revision of all my concerns. I have no additional questions or suggestions, and I recommend proceeding with the acceptance process according to the journal's guidelines.

**Do you want your identity to be public for this peer review?** For information about this choice, including consent withdrawal, please see our Privacy Policy

Reviewer #3: No

---

## [Editor Report · Acceptance letter]

PONE-D-25-09399R2

PLOS ONE

Dear Dr. Shim,

I'm pleased to inform you that your manuscript has been deemed suitable for publication in PLOS ONE. Congratulations! Your manuscript is now being handed over to our production team.

Kind regards,

on behalf of

Dr. Xing-Xiong An

Academic Editor

PLOS ONE